# Functional Characterization of Heat Shock Factor (*CrHsf*) Families Provide Comprehensive Insight into the Adaptive Mechanisms of *Canavalia rosea* (Sw.) DC. to Tropical Coral Islands

**DOI:** 10.3390/ijms232012357

**Published:** 2022-10-15

**Authors:** Mei Zhang, Zhengfeng Wang, Shuguang Jian

**Affiliations:** 1Guangdong Provincial Key Laboratory of Applied Botany, South China Botanical Garden, Chinese Academy of Sciences, Guangzhou 510650, China; 2Key Laboratory of South China Agricultural Plant Molecular Analysis and Genetic Improvement, South China Botanical Garden, Chinese Academy of Sciences, Guangzhou 510650, China; 3Key Laboratory of Vegetation Restoration and Management of Degraded Ecosystems, South China Botanical Garden, Chinese Academy of Sciences, Guangzhou 510650, China; 4Key Laboratory of Carbon Sequestration in Terrestrial Ecosystem, South China Botanical Garden, Chinese Academy of Sciences, Guangzhou 510650, China

**Keywords:** heat shock transcription factor, abiotic stress, ecological adaptation, *Canavalia rosea* (Sw.) DC

## Abstract

Heat shock transcription factors (Hsfs) are key regulators in plant heat stress response, and therefore, they play vital roles in signal transduction pathways in response to environmental stresses, as well as in plant growth and development. *Canavalia rosea* (Sw.) DC. is an extremophile halophyte with good adaptability to high temperature and salt-drought tolerance, and it can be used as a pioneer species for ecological reconstruction on tropical coral islands. To date, very little is known regarding the functions of *Hsf*s in the adaptation mechanisms of plant species with specialized habitats, especially in tropical leguminous halophytes. In this study, a genome-wide analysis was performed to identify all the *Hsf*s in *C. rosea* based on whole-genome sequencing information. The chromosomal location, protein domain or motif organization, and phylogenetic relationships of 28 *CrHsf*s were analyzed. Promoter analyses indicated that the expression levels of different *CrHsf*s were precisely regulated. The expression patterns also revealed clear transcriptional changes among different *C. rosea* tissues, indicating that the regulation of *CrHsf* expression varied among organs in a developmental or tissue-specific manner. Furthermore, the expression levels of most *CrHsf*s in response to environmental conditions or abiotic stresses also implied a possible positive regulatory role of this gene family under abiotic stresses, and suggested roles in adaptation to specialized habitats such as tropical coral islands. In addition, some *CrHsfA*s were cloned and their possible roles in abiotic stress tolerance were functionally characterized using a yeast expression system. The *CrHsfA*s significantly enhanced yeast survival under thermal and oxidative stress challenges. Our results contribute to a better understanding of the plant *Hsf* gene family and provide a basis for further study of *CrHsf* functions in environmental thermotolerance. Our results also provide valuable information on the evolutionary relationships among *CrHsf* genes and the functional characteristics of the gene family. These findings are beneficial for further research on the natural ecological adaptability of *C. rosea* to tropical environments.

## 1. Introduction

*Canavalia rosea* (Sw.) DC. (Fabaceae) is one of the representative species of pantropical plants with sea-drifted seeds, which has an extremely wide range of distribution in the tropical and subtropical seashore regions [1]. As an extremophile halophyte, *C. rosea* also shows strong tolerance to salt/alkaline, drought, high temperature, and barren soil and covers well with the purpose of wind-breaking and sand-fixing, and thereby, it has become a pioneer species for ecological reconstruction on tropical coral islands [2]. In particular, as compared with other leguminous plants, *C. rosea* presents great performance in adaptability to high temperature and it is more suited to tropical coral islands than most of other plant species in such environmental adversity [3]. Understanding the molecular and evolutionary mechanisms of *C. rosea*’s adaptation to these types of specialized habitats, especially to heat stress, would help to illuminate this extremophile’s adaptation to adverse conditions.

It has been reported that heat shock factor (Hsf) families are involved in plant complex signaling systems that mediate responses to high temperatures and also to a number of abiotic stresses such as cold, drought, hypoxic conditions, soil salinity, toxic minerals, strong irradiation, and even to pathogen threats [4]. The survival, growth, and reproduction of plants are significantly affected by some adverse abiotic stresses, which mainly include salinity, alkaline stress, drought, and temperature stress. Among these stresses, extreme hot weather has dramatically increased and high temperature stress has become one of the most common critical limiting factors that can remarkably suppress plant growth and affect the distribution of plants, leading to a significant reduction in crop yields and accompanied by a remarkable increase in global temperature [5,6]. Heat stress often affects the native state of cellular matrix, and therefore, causes folding, denaturation, or aggregation of proteins. Heat stress can also destroy cell membrane fluidity, cytoskeletal organization, change osmotic conditions and ion composition, and restrict transport and enzymatic reactions, thereby, resulting in metabolic imbalances and reactive oxygen species (ROS) outburst [7,8]. Plants have developed sophisticated protective mechanisms which can attenuate the damaging effects of environmental stresses. In response to heat stress, some specific response mechanisms are activated to maintain cellular homeostasis, regular growth, and metabolism, thereby, protecting the plant from environmental damage. Hsfs are an integral part of transcription controlling systems that regulate the activity of protective genes [4].

*Hsf* genes are extensively present in eukaryotes; however, the gene copies distinguish between the Plantae and Microbe/Animalia, and plants often possess large *Hsf* families with dozens of members [4], probably because they possess sessile features and need more complex regulatory networks. Basically, all plant Hsf members share conserved molecular structures. At the N-terminal, a DNA-binding domain (DBD), which is generally responsible for recognition of heat shock elements (HSEs, 5′-nGAAnnTTCn-3′) upstream of the TATA box in promoter regions, is characterized by a central helix-turn-helix motif and the most conserved part of Hsf proteins. The oligomerization domain (OD), following the DBD and near the N-terminal, is responsible for protein–protein interactions during transcriptional activation and are also involved in nuclear import and export [9]. Most Hsf members possess nuclear localization signal (NLS) and the nuclear export signal (NES) at the C-terminus, which function in the assembly of a nuclear import complex and the receptor-mediated export complex [4,10]. Generally, plant Hsfs are grouped into three classes, HsfA, B, and C [11]. Class A Hsf members also contain a unique activation domain (AHA domain, abbreviation for aromatic “W/F/Y”, larger hydrophobic “L/I/V”, and acidic “E/D” amino acid residues) motifs close to the NES, which are essential for transactivation [9], and Class B Hsf members contain a highly conserved tetrapeptide “-LFGV-”, which functions as a repressor domain instead of the AHA motif in Class A Hsf members [12]. Class C Hsf members do not possess the AHA motif or the repressor domain “-LFGV-” at their C-terminus, and their precise functions are not yet known.

A number of plant *Hsf* families from various species have been isolated and comprehensively studied, and increasing reports have indicated that *Hsfs* are involved in plant growth and development, as well as in responding to a number of abiotic stress responses, such as heat, drought, salinity, cold, heavy metal, and photo-oxidative stresses [4,10,12], especially by mediating the transcriptional regulation of heat shock proteins (HSPs), other chaperones, ROS scavengers and protective enzymes, and even some transcription factors [4,12]. So far, studies on plant *Hsf*s have centered on their functional versatility with the final purpose of designing novel strategies to use *Hsf*s for improving tolerance and adaptation of crops to adverse environmental conditions. Overall, as positive responding genes to stresses, Class A *Hsf*s have been well studied in many species, and their functions have been described as being involved in heat, drought, salt, oxidative damage responses, etc. [13,14]. However, much less is known about Class B and C *Hsf* members; only several previous reports have indicated that these genes may take part in salt/drought tolerance or thermotolerance [4,12,15,16].

The *C. rosea* genome sequencing project completed the full genome assembly, which provides important reference data for the functional identification of *C. rosea* genes and for the exploration of natural ecological adaptability to the specialized habitats of tropical coral islands. Among pantropical plants with sea-drifted seeds [2,17], *C. rosea* shows much better heat tolerance than most of other leguminous species. Investigations regarding the molecular mechanisms of heat tolerance would provide a foundation for further study on the adaptability of *C. rosea* or other plants to tropical coral islands for environmental improvements. Furthermore, *C. rosea* also shows great advantages associated with its tolerance of abiotic stresses such as high salinity and alkaline stress, drought, and nutrition deficiency. The *Hsf* gene family has been fully characterized only in Arabidopsis and some crop species, and much less is known about possible roles of this family for abiotic stress tolerance adaptation in specialized habitat species. In this study, a genome-wide identification of the bay bean *Hsf* family members was performed, and gene expression analyses were summarized. A total of 28 *CrHsf* members were identified in *C. rosea* using bioinformatics analysis and their expression levels were determined with RNA-seq analyses and quantitative reverse transcription PCR (qRT-PCR). Furthermore, we also performed functional identification of several *CrHsfA*s using a yeast heterogeneous expression system, which provided a basis for further function research on heat, salt/drought, and oxidative stress tolerant genes in *C. rosea*. The results in this study provide an important foundation to better understand the functional and evolutionary history of the *CrHsf* family in plant species, and also to help reveal the possible molecular mechanisms of *CrHsf*s involved in tolerance to various abiotic stresses and ecological adaptability to tropical coral island specialized habitats.

## 2. Results

### 2.1. Identification and Characterization of the C. rosea Hsf Family

A total of 28 putative *Hsf* genes were identified from the *C. rosea* genome. The CrHsf*s* were subdivided into three subfamilies, i.e., A, B, and C, according to their conserved regions or domains, including 15 CrHsfAs, 12 CrHsfBs, and 1 CrHsfC (Table 1). All of the *CrHsf* genes were named, in turn, according to their chromosome loci. Then, the biochemical characteristics of the Hsf proteins were predicted using the ExPASy proteomics server and Wolf PSORT program. The length of the *CrHsf*s’ coding DNA sequences (CDSs) ranged from 588 bp (*CrHsf5*) to 1488 bp (*CrHsf26*) with 195–495 amino acid residues. The molecular weights (Mws) of the CrHsfs varied significantly from 22.39 kilo dalton (kDa) (CrHsf5) to 54.79 kDa (CrHsf26), the predicted theoretical isoelectric points (PIs) ranged from 4.76 (CrHsf10) to 8.98 (CrHsf13 and CrHsf25), and the grand average of hydropathicity (GRAVY) values of all CrHsf proteins were negative, with the values between –0.928 (CrHsf25) and –0.517 (CrHsf24). Most of the CrHsfs (27) presented higher instability index values (II > 40) (except CrHsf25), indicating that these proteins acting as transcription factors (TFs) might be unstable in vivo, which also matched the characteristics of CrHsfs being involved in some post-translational modification (PTM) pathways with the purpose of regulating Hsf protein abundance by chaperone-mediated ubiquitination and degradation by the ubiquitin proteasome system (UPS) [10]. Additionally, subcellular localization predictions indicated that most CrHsf proteins were predicted to be located mainly in the nuclei, except that CrHsf9 was mainly targeted to cytoplasm and other organelles. The predicted results for the protein characteristics and subcellular localization of all CrHsfs are listed in Table 1.

### 2.2. Phylogenetic Analysis, Classification, and Conserved Motif Analyses of CrHsf Proteins

The CrHsf family members of the *Arabidopsis thaliana* (21 AtHsfs), *Zea mays* (25 ZmHsfs), *Glycine max* (38 GmHsf), *Cicer arietinum* (22 CaHsfs), and *Vigna radiata* (24 VrHsfs) were downloaded from the phytozome database. These protein sequences and 28 CrHsfs were analyzed by generating a neighbor-joining phylogenetic tree (Figure 1). *Canavalia rosea*, *G. max*, *C. arietinum*, and *V. radiata* all belong to leguminous plants, while *A. thaliana* and *Z. mays* are dicotyledonous and monocotyledonous model plants, respectively. CrHsfs, in general, showed higher homology with Hsfs from other leguminous plants, especially with GmHsfs from soybean (Figure 1).

The conserved domain of DBD (PF00447) is observed in all of the CrHsf proteins, while the HR-A/B domain is absent from the four CrHsfB4 subclass members (Table 2). The NLS and NES characteristic amino acids were detected with cNLS Mapper and NetNES programs, and the results indicated that not all CrHsfs had the typical nuclear localization signal and nuclear export signal (Table 2), which was consistent with the prediction for CrHsf proteins’ subcellular localization (Table 1); this result also indicated that the regulatory patterns of CrHsf proteins’ distribution were variable in vivo.

Based on the well-established Arabidopsis Hsf family classifications [20], as well as according to the HR-A/B region (or oligomerization domain, OD), the differences of the DBD amino acid sequences, and the linker length between HR-A/B regions and DBD domains, the 28 CrHsfs were classified into three subclasses: A, B, and C. Subclass A was divided into eight subclusters, designated as A1, A2, A4, A5, A6, A7, A8, and A9. Subclass B was divided into five subclusters B1, B2, B3, B4, and B5. Subclass C only contained one member, CrHsf22 (Figure 1 and Figure 2). There were no Hsf A3 members found in the *C. rosea* genome, which is quite different from other plant species, especially the leguminous plants. Since there are three GmHsfA3 members in soybean, two VrHsfA3 members in mung bean, and one CaHsfA3 member in chickpea (Figure 1), this result probably can be attributed to incomplete genome splicing and annotation of the *C. rosea* whole genome sequencing data assembly. The single copy subclass C member in *C. rosea* is similar with that in other leguminous plants. Instead, in many monocotyledonous plants, the subclass C Hsf C members are usually in multicopy [21,22,23,24].

In addition to the identification of these typical conserved domains of plant Hsfs shown in Table 2, we also detected the putative motifs using the Multiple Em for Motif Elicitation (MEME). A total of 10 different motifs were identified in CrHsfs with lengths ranging from 20 to 50 aa (Figure 2A). The conserved Motifs 1, 2, and 3 exist in most CrHsfs, except the B5 member, CrHsf5 (only containing Motifs 1 and 3). Motifs 4 and 5 only exist in Class A, while Motif 6 only presents in Class B CrHsfs. Motif 8 only exists in the A4 and A5 subgroups (Figure 2A). The motif composition and distribution of the same subgroup members is similar, but they showed significant variability among different subgroup members. We also drew the typical conserved domains (including DBD, OD, AHA, and RD) manually in the whole CrHsf family, which also showed similar structural composition with the MEME predicted result (Figure 2B).

### 2.3. Chromosomal Location and Duplication of CrHsf Genes

There are, in total, eleven chromosomes in the *C. rosea* genome, while each chromosome has different numbers of *CrHsf* genes (Figure 3A). Chromosomes 1 and 4 have the most *CrHsf* genes (five genes on each chromosome), followed by Chromosome 10 (harboring four genes). Chromosomes 5 and 6 both have three *CrHsf* genes, and Chromosomes 2, 3, 7, and 11 all contain two *CrHsf* genes. No *CrHsf* genes were located on Chromosomes 8 and 9.

The gene duplication events also reflect the evolutionary relationship of different gene members in the family, which are also considered to be important mechanisms for plant adaptive evolution at the genome level. The homology analysis showed that there was no tandem duplication event in the *C. rosea CrHsf* family, and a total of ten *CrHsf* gene pairs were found to be segmentally duplicated (Table 3 and Figure 3B). The selection pressure acting on *CrHsf* genes was inferred from the ratio of non-synonymous (Ka) to synonymous (Ks) substitution values. Our data indicated that all *CrHsf* genes were under evolutionary pressure, with an average Ka/Ks ratio of 0.2194 (Table 3). The Ka/Ks values of the gene pairs were all far lower than 1.0, which suggested that these gene pairs were mainly selected for purification during the evolution process with limited functional divergence after duplication.

### 2.4. Gene Structures and Cis-Acting Element Analyses in CrHsf Promoter Regions

The gene structural differences also inflect the evolutionary relationship of gene members in the same family. The exon/intron structures within a gene family showed differences and similarities to some degree. In this study, the structural differences of the *CrHsf* genes were also analyzed, mainly based on the exon/intron frames. Most *CrHsf* genes possessed only one intron, and only *CrHsf9* and *CrHsf13* had two introns (Figure 4). In general, the *CrHsf* family illustrates a very highly conserved exon/intron splicing arrangement in the *C. rosea* genome, and in the same subgroup, the length of exon is similar, while the intron length exhibits more differences.

The *cis*-acting elements in promoter regions are the key factors that affect the expression of genes in a cellular-specific manner or responding to different environmental challenges. In order to explore the potential functions of *CrHsf*s, the *cis*-acting elements within the promoters of the 28 *CrHsf*s were analyzed using PlantCARE, assisted with manual characterization of some special elements, such as HSEs. The 2 kb DNA sequences (possible promoter regions) before each transcriptional start site (TSS) were predicted for all 28 *CrHsf*s in the *C. rosea* genome (Figure 5). The statistical *cis*-acting elements include several hormone responsive elements, such as MeJA-responsive, auxin-responsive, GA-responsive, SA-responsive, ethylene-responsive elements, and ABRE (abscisic acid-responsive element), as well as other typical stress or development-related elements, including anaerobic-responsive (flooding-related responsive), TC-rich repeats (involved in defense and stress responsive), MYC (MYC binding site), MYB (MYB binding site), as-1 (stress responsive), LTR (abiotic stress responsive) elements, and HSEs (heat shock elements) (Figure 5A). These elements are displayed in Figure 5. Almost all of these genes’ promoter regions held multiple ABREs, MYCs, MYBs, TC-rich repeats, and HSEs, suggesting that their expressions were responsive to multiple abiotic stresses, possibly including drought, salt/alkaline stress, and heat challenges. In general, the promoter analysis suggested that all of the *CrHsf*s should be regulated precisely at the transcriptional level and respond to various environmental stimulation.

Combined with our predicted potential interactions, among the CrHsf proteins with the soybean database as reference (Appendix A), the protein interaction network mediated by CrHsfs seemed to be much complicated, and some CrHsfs could be directly or indirectly bound or interacted with each other or some molecular chaperones, such as heat shock proteins (HSPs). Given the fact that plant homomeric and heteromeric Hsf–Hsf interactions are pretty ubiquitous, the plant Hsf protein combinations seem to be more complicated [4] and further indicate that the expressions of *CrHsf*s are also alternatively regulated by different CrHsf protein complexes, which also explains the high frequency of HSEs in *CrHsf*s’ promoter regions (Figure 5B).

### 2.5. Expression Profiles of CrHsf Genes in Different Tissues or in Response to Different Habitats

To understand the tissue-specific expression patterns of *CrHsf*s in *C. rosea*, root, stem, leaf, flower bud, and young fruit tissues were used to quantify their expressions under normal conditions (growing in the SCBG) (Figure 6A). The expressions of most *CrHsf*s were lower in leaf than those in root, stem, flower, and fruit under non-stressful environment conditions, although *C. rosea* leaf is the main and direct organ for sensing heat or highlighting stress.

To explore the possible roles of *CrHsf*s for adaptation to tropical coral islands, we also investigated the expression differences of *CrHsf* genes in adult *C. rosea* plants growing on Yongxing Island (YX sample) and in the South China Botanical Garden (SCBG sample). The mature leaves were collected and detected via RNA sequencing technique. According to the FPKM values, the expression profiles of the *CrHsf*s differed considerably in the two samples (Figure 6B). Except for several *CrHsf*s (*CrHsf13*, *CrHsf15*, *CrHsf16*, and *CrHsf22*) with relatively stable expression patterns, some *CrHsf*s showed obviously elevated expression levels in the YX sample as compared with those in the SCBG sample, including *CrHsf2*, *CrHsf7*, *CrHsf9*, *CrHsf17*, *CrHsf21*, *CrHsf24*, *CrHsf25*, and *CrHsf2**8*, while some *CrHsf*s, such as *CrHsf1*, *CrHsf10*, *CrHsf12*, *CrHsf14*, *CrHsf23*, and *CrHsf26,* showed slightly downregulated expression patterns in the YX sample. The results suggested the involvement of *CrHsf*s responding to environmental adversity, especially, *CrHsf*s playing positive roles in native *C. rosea* plants growing on tropical coral islands. In addition, the results further verified that the *C**rHsf*s were involved in ecological adaptability against multiple abiotic stresses in *C. rosea* plants’ native habitats.

### 2.6. Expression Profiles of CrHsf Genes in Response to Different Abiotic Stresses

The major role of *Hsf* genes is to respond to heat challenge, and the primary limitation factor for vegetation growth on tropical islands is the constant extreme high temperature. To explore the responses of *CrHsf*s to heat stress, we analyzed the expression profiles of *CrHsf*s in leaves and roots of *C. rosea* seedling plants placed in a 45 °C thermostatic light incubator for two hours. As shown in Figure 7A, both in root and in leaf samples, *CrHsf**4*, *CrHsf7*, *CrHsf9*, *CrHsf10*, *CrHsf11*, *CrHsf13*, *CrHsf16*, *CrHsf17*, *CrHsf19*, *CrHsf24*, and *CrHsf25* were significantly induced by heat treatment, while *CrHsf8*, *CrHsf15*, *CrHsf**20*, *CrHsf**23*, and *CrHsf2**7* showed downregulate expression patterns under heat challenge.

To extend our understanding of *CrHsf* responses to other stresses in *C. rosea* plants growing on tropical islands, including high salinity, alkaline toxicity, and drought, we mimicked these adversities with NaCl, NaHCO_3_, and mannitol solutions to challenge the *C. rosea* seedlings. As shown in Figure 7B,C, in general, the changes in the expression levels of *CrHsf*s both in roots and in leaves caused by high salinity, alkaline stress, and osmotic stress were relatively smaller than those caused by heat stress, although several *CrHsf* members, such as *CrHsf7*, *CrHsf8*, *CrHsf16*, *CrHsf17*, *CrHsf22*, *CrHsf24*, and *CrHsf25*, expressed higher under challenges than in normal conditions.

We also performed qRT-PCR to confirm the expression patterns of 10 selected *CrHsf*s (*CrHsf2*, *CrHsf7*, *CrHsf9*, *CrHsf10*, *CrHsf16*, *CrHsf17*, *CrHsf2**1*, *CrHsf**24*, *CrHsf25*, and *CrHsf28*) in salt, alkaline, high osmotic, and heat stresses, mainly based on their RNA-seq data. The results illustrated that almost all of the selected *CrHsf*s showed similar expression patterns under the same stress conditions as compared with the RNA-seq results. Alkaline stress, as a result of higher environmental pH, usually triggers more severe damage to plants than simple saline stress with a neutral pH; therefore, plants must respond quickly to alkaline stress. As shown in the results (Figure 8), under the alkaline and heat stress treatments, the expression of all the 10 *CrHsf*s increased rapidly and significantly (Figure 8). These results may indicate that the induced changes in the expression levels of *CrHsf*s caused by heat are quick and vigorous, while the effect of alkaline stress is also much stronger than that due to salt and high osmotic stresses.

### 2.7. Transactivation Activity Analysis of the CrHsf Proteins

The transcriptional activities of some candidate CrHsf proteins were examined using a yeast expression system. The fusion plasmids CrHsfs-pGBKT7-BD and pGBKT7-BD (negative control) were transformed into yeast strain AH109, and grown on an SD medium lacking tryptophan (SD/Trp-) or lacking tryptophan and histidine (SD/Trp-His-). The growth status of transformants was evaluated (Figure 9). Yeast cells containing either pGBKT7 (GAL4-BD) or CrHsfs-pGBKT7 (GAL4-BD-CrHsfs) grew well on the SD/Trp- plates; however, only cells containing CrHsfs-pGBKT7 grew on the SD/Trp-His- plates and turned blue in the presence of 5-bromo-4-chloro-3-indoxyl-α-D-galacto-pyranoside (X-α-gal), which showed that *LacZ*, the second reporter gene, was activated by these eight CrHsfs (Figure 9). The positive control CrASR1-pGBKT7-BD also grew well on the SD/Trp-His- plate and turned blue with X-α-gal as a substrate, since CrASR1 has been proven to be a transcription factor [25]. The above results demonstrated the presence of transcriptional activities in all of cloned CrHsf proteins.

Plant HSEs (5′-GAAnnTTC-3′) are recognized and bound by specific Hsf TFs. Here, we performed a dual-luciferase (LUC) assay to investigate the effect of several *CrHsf*s’ overexpression on the activities of HSEs and mutated HSEs (mHSEs, 5′-GACACACT-3′). Without exception, the overexpression of all eight *CrHsf*s significantly increased the LUC activity of HSE-pGreenII0800-LUC, but had no significant effect on that of mHSE-pGreenII0800-LUC in tobacco leaf cells (Figure 10). These results indicated that CrHsfs showed transcription activation activities and could act as transcription factors to activate the expression of target genes in which their promoter regions contain HSE sequences.

### 2.8. Heterologous Expression of CrHsfs Confers Abiotic Tolerance in Transgenic Yeast

To identify the potential roles of CrHsfs in vivo, we performed a series of heterogeneous expression assays of the above mentioned eight *CrHsf*s (*CrHsf1*, *CrHsf2*, *CrHsf7*, *CrHsf9*, *CrHsf10*, *CrHsf15*, *CrHsf17*, and *CrHsf23*) with a yeast system for a functional stress tolerance investigation. Firstly, for the antioxidation tolerance test, the eight *CrHsf*s-pYES2 were introduced into two H_2_O_2_-sensitive mutant strains *skn7Δ* and *yap1Δ*, with the corresponding wild-type (WT) yeast BY4741 and two mutant strains transformed with the empty vector pYES2 as controls. The *skn7Δ*, *yap1Δ*, and WT strains, harboring different *CrHsf*s-pYES2, were also used to test the thermotolerance of yeast possibly mediated by *CrHsf*s. In brief, all eight *CrHsf*-pYES2s showed varying degrees of increased tolerance to H_2_O_2_ both in *skn7Δ* (Figure 11A) and in *yap1Δ* (Figure 11B). This result indicated that these *CrHsf*s all possessed some antioxidation activities when overexpressed in yeast cells, probably by inducing the expression of some antioxidant genes. We also tested the thermotolerance of *skn7Δ* and *yap1Δ* strains harboring *CrHsf*s-pYES2 with WT harboring empty vector pYES2 as control. As shown in Figure 11, all the tested *CrHsf*s could increase the thermotolerance of *skn7Δ* and *yap1Δ* strains to different degrees, while the *skn7Δ* and *yap1Δ* with empty vector pYES2 showed almost lethal phenotype after 52 °C heat challenge for 15 min. In addition, the WT also presented regular growth condition after this type of heat stress. Further, we extended the heat challenge time (52 °C 30 min) with yeast WT strain harboring *CrHsf*s-pYES2 and pYES2, and we found similar results (Figure 12), that is, the WT yeast expressing different *CrHsf*s also showed obviously improved thermotolerance as compared with that harboring empty vector pYES2, which confirmed these *CrHsf*s could increase the tolerance both to oxidative stress and to heat. The expressions of all eight *CrHsf*s in WT yeast were also assessed with NaCl and sorbitol treatments, and our results indicated that none of *CrHsf*s could increase the salt and high osmotic stress tolerance of yeast when expressed in the WT strain (Appendix A). The yeast stress tolerance results are preliminary, and there is still a need for further functional identification in plants using transgenic assays.

## 3. Discussion

The tropical coral islands often possess the characteristics of high temperature, high alkaline stress, high salinity, high light, freshwater shortage, and soil fertility shortage (abbreviated in “four-high and two-shortage”) for vegetation growth, thereby being barren for most plant species. The ecological structure of such islands is simple and exceedingly fragile. *Canavalia rosea* (Sw.) DC., which is a sea floating halophyte, has strong tolerance to salt/alkaline, drought, heat, and barren soil and covers well with the purpose of wind-breaking and sand-fixing, and thereby, has become a pioneer species for ecological reconstruction on tropical coral islands. Among all of the ecological factors that determine the distribution of plants in tropical regions, high temperature (heat stress) and related hydrothermal conditions are key environmental factors that control the specialized habitat vegetation distribution and adaptability to the physical environment, the soil composition, and the sea salinity. Plant Hsfs are a class of TFs that have been proven to play critical regulatory roles in plant responses to various biotic or abiotic stresses, especially to heat stress. Clearly, it can be inferred that in the *C. rosea* plant, the *CrHsf* family could be involved in the molecular mechanisms or pathways underlying this extremophile halophyte’s adaptation to tropical native habitats and its responses to acute heat stress, high salinity/alkaline, and even water-deficit conditions.

In this study, a genome-wide survey was carried out in *C. rosea* based on its genome sequencing data, and we characterized 28 *CrHsf* genes in total. The corresponding CrHsf proteins were phylogenetically clustered into three subfamilies, including 15 CrHsfAs, 12 CrHsfBs, and 1 CrHsfC. The *Hsf* gene families have already been identified and functionally characterized in several leguminous plants, including *Glycine max*, *Lotus japonicus*, *Medicago truncatula*, *Cicer arietinum*, and *Vigna radiata* [18,19,26,27,28], and in recent years, the *Hsf* families have also been extensively explored in the developmental regulation and response to abiotic stresses in some cash crops, such as *Phyllostachys edulis* [23], *Ananas comosus* [24], *Camellia sinensis* [29], *Lactuca sativa* [30], *Hypericum perforatum* [31], *Prunus persica* [32], and *Gossypium barbadense* [33]. In specialized habitat plant species, especially some native halophytes on tropical islands or the coast with strong adversity resistance, it has been suggested that the *Hsf* family has been involved in adaptive evolution in which plants promote their adaptability and improve their survival under extreme heat, high salinity/alkaline stress, drought, or highlight/UV exposure environments by altering their DNA structure or changing their gene expression levels [19,30]. *Canavalia rosea*, which is a leguminous halophyte, shows great advantages in heat tolerance and saline-alkaline/drought resistance. Certainly, *C. rosea* has developed elaborate mechanisms for adapting to the multiple adversity environments on tropical coral islands or other coastal zones, and thus, the identification of the stress relevant *CrHsf* family in this species should help to clarify the molecular mechanisms of plants’ adaptive evolution to extreme adversity, especially to high temperature challenges.

As compared with a single *Hsf1* in *Saccharomyces cerevisiae* and *Drosophila melanogaster*, or with only seven *Hsf* members in humans [34,35], the *Hsf* family in plants is relatively large while variable. There are 27 *Hsf*s in Arabidopsis (*Arabidopsis thaliana*) [11], 31 in maize (*Zea mays*) [21], 27 in rice (*Oryza sativa*) [36], 30 in common bean (*Phaseolus vulgaris*) [37,38], 24 in mung bean (*Vigna radiata*) [27], 22 in chickpea (*Cicer arietinum*) [19], 22 in pineapple (*Ananas comosus*) [24], and 23 in petunia (*Petunia hybrida*) [39]. In general, the number of 28 *Hsf* genes in *C. rosea* genome is intermediate (Table 1). Nevertheless, there are still some possible events of gene duplication among the *CrHsf* genes (Table 3). Gene duplication usually contributes to the extension of gene families in plant genomes, which have been considered to be the result of evolutionary adaptation to natural environmental changes. In this study, gene duplication analyses provided additional information regarding the evolution of *CrHsf* genes. We found that both subfamily A and subfamily B held similar numbers of segmentally duplicated gene pairs (Table 3), and in the B4 subclass, there were two pairs of duplicated gene pairs (*CrHsf6*/*CrHsf20* and *CrHsf18*/*CrHsf27*). HsfB factors are considered to be repressors of heat shock responses, mainly modulating the action of class HsfAs by forming heterotrimers with HsfAs through their OD motifs [4]. Here, we supposed that the more expansion in *CrHsfB* (five pairs duplications corresponding to 12 members) than that in *CrHsfA* (five pairs duplications corresponding to 15 members) was probably due to the more precise regulatory mechanisms when under heat challenges in *C. rosea* plants. In addition, as compared with other diploid temperate leguminous crops such as mung bean (10 *VrHsfB*s in a total of 24 *VrHsf*s, 41.7%) [27], chickpea (nine *CaHsfB*s in a total of 22 *CaHsf*s, 40.9%) [19], and common bean (12 *PvHsfB*s in a total of 30 Pv*Hsf*s, 40%) [37], 12 *CrHsfB*s in a total of 28 *CrHsf*s (42.8%) may also be a type of indicator of evolutionary fitness by B class *CrHsf*s’ gene expansion for regulating multiple stress responses in *C. rosea*, including thermotolerance and other environmental adversities. A soybean class B heat shock factor, GmHSFB2b (Glyma.11G025700.1, GmHsf-26), has been proven to be involved in improving soybean’s salt tolerance through activating flavonoid biosynthesis and accumulation; also, in salt-tolerant wild and cultivated soybean lines (Y20 and Y55), the promoters of this gene have shown higher activities under salt stress than that in salt-sensitive accession soybean line (Y0523) [40]. This is the first report that plant HsfBs function as transcriptional regulatory factors that play an important role in salt stress response during domestication. Here, our result about the *CrHsf* family might also be ascribed to the ecological strategies associated with *C. rosea*’s environmental adaptation, resulting in the genes’ expansion and specialized *CrHsf*s’ gene functions in same subclass.

Being a type of TFs, Hsfs are highly evolutionarily conserved in all eukaryotes and regulate various stress-responsive and biological process-related genes in plants [4]. In this study, despite their variable sequences of different subfamilies (A, B, and C), all the CrHsfs present a remarkable conserved structure, that is, the N-terminal DNA-binding domain (DBD) (Table 2 and Figure 2). Most CrHsfs also contain the oligomerization domain (OD) adjacent to DBD and the C-terminal activator peptide motif (AHA), while the repression domain (RD) is present towards the C-terminus and is a characteristic of class B CrHsfs (except B5 member CrHsf5) (Figure 2B). Correspondingly, the presence of nuclear localization signal (NLS) and leucine-rich nuclear export signal (NES) sequences would ensure that the shuttles of CrHsfs among the nuclei, nucleoplasm and cytoplasm are TFs and stress-responsive proteins (Table 2). The conservation and specificity of different subclasses of CrHsfs can provide different stress responses or developmental issues with specific members, meanwhile, they can also provide the functional collaboration or redundancy to deal with special or extreme circumstances.

Many previous studies have demonstrated that plant *Hsf* genes were involved in protecting plants from stresses, especially from heat challenges, as well as being involved in plant development and defense processes [12]. The regulation of *CrHsf*s’ expression must be a primary determinant of this gene family of stress-responsive genes that play central roles in the tolerance of *C. rosea* plants against complex abiotic stresses. The *cis*-acting elements located in *CrHsf*s’ promoters were systematically analyzed in this study. Most *CrHsf*s’ promoters contained multiple HSE, ABRE, MYB, MYC, and TC-rich repeat *cis*-elements (Figure 5), which suggested that *CrHsf*s could be involved in various stress responses, including heat, salt/alkaline, and drought. Specifically, the presence of a large amount of HSE in their promoter is further proof that the *CrHsf* family is largely self-regulated at the transcriptional level.

It is well known that the expression of *Hsf* genes is triggered by multiple stresses [12] and plays vital roles in a variety of abiotic stress responses and plant development processes [4]. For example, common bean *PvHsf*s were differentially expressed under cold, heat, salt, and heavy metal stresses [37], and several members, including *PvHSFA8*, *PvHSFB1A*, and *PvHSFB2A*, showed progressively induced patterns of varying degrees under heat stress conditions [38]. The transcriptome analysis of *Sorbus pohuashanensis* under heat, drought, and salt stresses showed that some of *SpHsf*s genes were generally induced by these abiotic factors, while some exhibited low expression levels in specific stages of the abiotic stress progression, indicating that *SpHsf*s played functional roles in abiotic stress responses in *S. pohuashanensis* [41]. For phosphate deficiency, the expression of wheat *TaHsfA2d* has been shown to be strongly repressed, and overexpression of *TaHsfA2d-4A* in Arabidopsis resulted in significantly enhanced sensitivity to Pi deficiency, indicating that wheat TaHsfA2d participated in the regulation of Pi deficiency stress [42]. In our study, the RNA-seq analysis showed that the *CrHsf* family presented a broad expression pattern across different organs and tissues at various developmental stages, while their expression appeared to be relatively low in mature leaf under regular growth conditions (Figure 6A). In addition, the expression profiles of *CrHsf*s in mature leaves captured from *C. rosea* plants growing in the SCBG (fit environment) or on Yongxing Island (harsh environment) exhibited huge differences (Figure 6B), which further indicated that *CrHsf*s might be widely involved in environmental adaptation to tropical coral islands. We further simulated the main limiting factors on tropical coral islands that mainly affect the growth of plants, including high salinity, alkaline stress, high osmotic stress (water-deficit), and high temperature, under which the *C. rosea* plant samples were collected and RNA-seq was performed. We observed very strong up-/downregulation of *CrHsf*s during heat stress (Figure 7A), while there seemed to be less change in the expression patterns under high salinity/alkaline and osmotic stresses (Figure 7B,C). In addition, our further qRT-PCR results concerning several target *CrHsf*s confirmed their transcriptional changes (Figure 8). Basically, the expression pattern of genes could be used as an indicator of their putative biological functions. The expression of apple *MdHSFA8a* is induced by high osmotic stresses and drought, and *MdHSFA8a* is involved in drought-induced accumulation of flavonoids and anthocyanins in transgenic apple plants, thus, mediating ROS scavenging under natural drought conditions [43]. *Moringa oleifera* is a drought-tolerant plant that is naturalized in tropical and subtropical regions around the world. *MolHSF8* has shown a significant upregulation in response to drought stress, indicating that it might be a promising candidate gene for functional characterization in drought tolerance in plants [44]. The tea plant (*Camellia sinensis*) *CsHsfA2* showed obviously induced expression pattern under heat stress, and heterologous expression of *CsHsfA2* conferred thermotolerance in transgenic yeast and Arabidopsis [45]. Similarly, the wheat (*Triticum aestivum*) *TaHsfA2-10* could be induced markedly by heat, and overexpression of *TaHsfA2-10* in Arabidopsis could clearly elevate the thermotolerance of transgenic plants [22]. It is clear that the induced expression of different *CrHsf*s in *C. rosea* plants also offers the opportunity for new functional identification for abiotic stress tolerance genes with *CrHsf*s as targets; related research also helps in elucidating the special molecular mechanisms that *C. rosea* plants adapt to its native habitats such as tropical coral islands.

The *Hsf* gene family is widespread in plants, and its members play important roles by regulating the expression of target genes such as HSPs. The regulation of target gene expression is the main function of the TF members. Here, we also detected the self-activation activities of eight CrHsfs using a yeast one-hybrid assay (Figure 9) and transcriptional activities using a dual-LUC assay (Figure 10). Without exception, all of detected CrHsfs exhibited the capability of self-activation, which is a key prerequisite to be TFs. In addition, the transcriptional activation of HSE in the dual-LUC assay confirmed that the eight CrHsfs were all transcription factors that could recognize and bind HSEs. A previous report indicated that the transactivation potential of plant Hsfs did not depend on the presence of C-terminal AHA motifs [46], which were only present in the Hsf A subfamily. Here, we chose eight CrHsfAs mainly based on their induced expression patterns according to the RNA-seq assays. Related studies have shown that tall fescue (*Festuca arundinacea*) *FaHsfA2c* acted as a positive regulator conferring thermotolerance by improving the expression of multiple HSPs both in tall fescue and in transgenic Arabidopsis [13]. Apple (*Malus domestica*) *MdHSFA8a* has been shown to participate in *MdHSP90* and *MdRAP2.12* mediating drought tolerance and activating the expression of ABA signaling-related genes, therefore, modulating the flavonoid synthesis to regulate drought tolerance [43]. In addition, citrus (*Citrus reticulata*) CitHsfA7 has shown transactivation effects on *CitAco3*, *CitIDH3,* and *CitGAD4* genes, and contributed to citric acid degradation in citrus fruit under hot air treatment (HAT) [47]. Our results indicate that these CrHsfs might enhance the heat and other abiotic stress tolerance of *C. rosea* plants by regulating or activating the expression of some target genes.

It has been shown that overexpressed *Hsf*s in plants could enhance their tolerance to heat or other abiotic stresses [13,22,43,45]. In order to further reveal the potential functions of the *CrHsf* genes, we performed rapid overexpression assays with a yeast system. Without exception, overexpression of *CrHsf*s in WT yeast enhanced heat tolerance (Figure 12), and a similar result has also been observed with wheat *TaHsfA2-1′*s overexpression in yeast [48]. ROS accumulation induced by abiotic stresses is harmful to plant growth and development. The mutation of *AtHsfA4a* could cause the higher content of H_2_O_2_ than that in WT plant under salinity stress [12]. The high expression level of *SaHsfA4*c could induce the expression of ROS-scavenging system-related genes (*POD*, *CAT,* and *APX*) in transgenic non-hyperaccumulation ecotype *Sedum alfredii* plants under Cd challenge, thereby, reducing reactive oxygen species (ROS) accumulation and Cd toxicity [49]. We also investigated the overexpression of *CrHsfs* in H_2_O_2_ sensitive mutant stain *skn7∆* and *yap1∆*, and the expressed *CrHsf*s could significantly improve the survival rate under H_2_O_2_ treatment as compared with that only transformed with pYES2, indicating that *CrHsf*s had been involved in environmental heat stress which could generate ROS. Similarly, the heat sensitivity of both *skn7∆* and *yap1∆* could be recovered to some extent by *CrHsf*s (Figure 11), which provided further evidence of CrHsfs acting as stress-associated proteins.

## 4. Materials and Methods

### 4.1. Plant Materials and Stress Treatments

*Canavalia rosea* adult plants growing in the South China Botanical Garden (SCBG, 23°18′76″ N, 113°37′02″ E) were transplanted gradually from Hainan Province, China, since 2012, and the *C. rosea* adult plants growing on Yongxing Island (YX, 16°83′93″ N, 112°34′00″ E) were also used in this study. The *C. rosea* seeds were also collected from coastal areas of Hainan Province. The seedlings of *C. rosea* were generated from seed germination and were grown, for 30 days, in a soil/vermiculite mixture with regular water and fertilizer supply, for further experiments.

To analyze tissue-specific transcriptional patterns of the identified *CrHsf*s, roots, stems, leaves, flower buds, and young fruits were gathered from *C. rosea* plants grown in the SCBG. In addition, to investigate the involvement of the *CrHsf*s in adaptation to different habitats, adult leaves were gathered from *C. rosea* plants growing on YX and in the SCBG. For abiotic stress treatments, the 30-day *C. rosea* seedlings were removed from the pots and carefully washed with distilled water to remove soil from the roots, following which they were transferred into 600 mM NaCl (1/2 Hoagland solution) for high salinity stress. For alkaline stress, the cleaned *C. rosea* seedlings were soaked in 150 mM NaHCO_3_ (pH 8.2) (1/2 Hoagland solution). For drought (high osmotic or water-deficit) treatment, the seedlings were soaked in 300 mM mannitol (1/2 Hoagland solution). For heat-shock treatment, the *C. rosea* seedlings were soaked in 45 °C prewarmed 1/2 Hoagland solution with the roots submerged and placed in a 45 °C thermostatic light incubator for two hours. All samples were immediately frozen in liquid nitrogen and stored at −80 °C for subsequent gene expression analysis. Three independent biological replicates were used. Illumina RNA sequencing was performed by the Shanghai OE Biotech Company (Shanghai, China), and the expression data (FPKM values) were visualized using TBtools.

### 4.2. Identification and Bioinformatics Analysis of the Hsf Family in C. rosea

The putative Hsf family members were identified by performing HMM searches against the *C. rosea* genome database by using an HSF_DNA-binding domain (PF00447); in order to confirm the accuracy of identified genes, protein sequences of the candidate *CrHsf* family members were submitted to SMART (http://smart.embl-heidelberg.de/, accessed on 1 May 2022) and the NCBI Conserved Domain Database (https://www.ncbi.nlm.nih.gov/Structure/cdd/wrpsb.cgi, accessed on 1 May 2022) to confirm the presence of HSF-type DNA-binding domain. Finally, the selected *CrHsf*s were named based on their gene loci in the *C. rosea* genome annotations. The obtained *CrHsf*s nucleotide and protein sequences from *C. rosea* are listed in Appendix A.

The CDS length, protein length, theoretical molecular weight (MW), isoelectric point (pI), instability index (II), and grand average of hydropathicity (GRAVY) value of each CrHsf were analyzed using the ProtParam program (http://web.expasy.org/protparam/, accessed on 1 May 2022). Subcellular localization of CrHsfs was predicted using the WoLF PSORT server (https://wolfpsort.hgc.jp/, accessed on 1 May 2022). The exon-intron structures of the *CrHsf* genes were analyzed using the web server GSDS (Gene Structure Display Server, http://gsds.cbi.pku.edu.cn/, accessed on 1 May 2022) based on the comparison of cDNA sequences with their genomic DNA sequences. CrHsf protein sequences were uploaded to the STRING database (https://string-db.org/, accessed on 1 May 2022) for node comparison, and relationships among important proteins were predicted based on soybean protein interactions.

### 4.3. Phylogenetic and Sequence Conservation Analysis of C. rosea

An unrooted neighbor-joining phylogenetic tree was created based on multiple protein sequence alignments of all identified CrHsfs from *C. rosea* using ClustalW and MEGA 6 with 1000 bootstrap replicates. To analysis the evolutionary relationship of plant Hsf members, the sequences of GmHsfs (*Glycine max*), AtHsfs (*Arabidopsis thaliana*), CaHsfs (*Cicer arietinum*), and ZmHsfs (*Zea mays*) were downloaded from the phytozome database (https://phytozome-next.jgi.doe.gov/, accessed on 1 May 2022), and VrHsfs from mung bean (*Vigna radiata*) were downloaded from the NCBI database (https://www.ncbi.nlm.nih.gov/, accessed on 1 May 2022) according to previous reports [18,19,21,26,27]. The circular neighbor-joining phylogenetic tree was constructed with MEGA 6.0 and embellished with WPS Office.

Conserved CrHsf motifs were analyzed using the MEME suite (http://meme-suite.org/, accessed on 1 May 2022), with the maximum number of motifs being 10 and the optimum width of motifs ranging from 11 to 50. The conserved CrHsf domains were also detected using the Pfam databases (http://pfam.xfam.org/, accessed on 1 May 2022) and SMART (http://smart.embl-heidelberg.de/, accessed on 1 May 2022) to confirm the HSF_DNA-binding (DBD) domain and oligomerization domain (OD or HR-A/B, heptad hydrophobic repeat A or B, containing a typical coiled-coil structure). The nuclear localization signal (NLS) and nuclear export signal (NES) were searched with cNLS Mapper (http://nls-mapper.iab.keio.ac.jp/cgi-bin/NLS_Mapper_form.cgi, score ≥ 5.0, accessed on 1 May 2022), and NetNES (http://www.cbs.dtu.dk/services/NetNES/, score > 0.5, accessed on 1 May 2022). The AHA and RD motifs were predicted manually according to previous reports [11,18,19]. The gene structure for each *CrHsf* was illustrated using the Gene Structure Display Server 2.0 (http://gsds.cbi.pku.edu.cn/, accessed on 1 May 2022).

### 4.4. Gene Duplication and Gene Collinearity Analysis

Gene segmental and tandem duplications were assessed using the MCScanX software (http://chibba.pgml.uga.edu/mcscan2/, accessed on 1 May 2022), and tandem duplications were also checked manually according to their gene loci. The number of synonymous substitutions per synonymous site (Ka), the number of non-synonymous substitutions per non-synonymous site (Ks), and the *p*-value from a Fisher’s exact test of neutrality were calculated using the Nei-Gojobori model with 1000 bootstrap replicates [50]. A Ka/Ks ratio < 1 indicated purifying selection, a Ka/Ks ratio = 1 indicated neutral selection, and a Ka/Ks ratio > 1 indicated positive selection.

### 4.5. Analysis of Cis-Acting Elements of CrHsf Promoters

Putative *Cr**Hsf* promoter sequences (2000 bp upstream of ATG) were retrieved from the *C. rosea* genome database (Appendix A). Sequences were then uploaded into the PlantCARE database (http://bioinformatics.psb.ugent.be/webtools/plantcare/html/, accessed on 1 May 2022) for *cis-*acting regulatory element analysis. The *cis*-acting elements were classified as either hormone-specific elements (gibberellin-responsive elements, MeJA-responsive elements, auxin-responsive elements, salicylic acid-responsive elements, EREs, and ABREs) or abiotic stress-responsive elements (anaerobic responsive elements, TC-rich repeats, MYC, MYB, as-1, LTRE, and HSE). The different elements were summarized in Appendix A and several selected *CrHsf* promoters were visualized using TBtools [51].

### 4.6. Expression Profile Analysis of CrHsfs and Other Abiotic-Related Genes in Various Tissues/Organs or under Multiple Stresses

RNA-seq analyses of different *C. rosea* tissues with or without stress challenges were performed to obtain the differential expression gene (DEG) data. In brief, the *C. rosea* RNA-seq datasets were constructed using Illumina HiSeq X sequencing technology with the FastQC program (http://www.bioinformatics.babraham.ac.uk/projects/fastqc/, accessed on 1 May 2022) based on the primary 40 Gb clean reads and were mapped to the *C. rosea* reference genome using Tophat v.2.0.10 (http://tophat.cbcb.umd.edu/, accessed on 1 May 2022). Gene expression levels were calculated as fragments per kilobase (kb) of transcript per million mapped reads (FPKM) according to the length of the gene and the read counts mapped to the gene: FPKM = total exon fragments / [mapped reads (millions) × exon length (kb)]. Expression levels of *Cr**Hsf*s were visualized as clustered heatmaps (log2) using TBtools, which were directly shown with FPKM values. The FPKM values of *CrHsf*s for RNA-Seq assays were summarized in Appendix A.

A qRT-PCR was carried out to further confirm the expression patterns of *Cr**Hsf*s in *C. rosea* plants under abiotic stresses, with the same treatment conditions as that in the RNA-seq assay. Total RNA of the root, vine, and leaf samples were extracted separately using a plant RNA extraction kit (TransGen Biotech, Beijing, China), according to the manufacturer’s instructions. RNA concentration and quality were tested using a NanoDrop 1000 (Thermo Fisher Scientific, Waltham, MA, USA), with the integrity checked on 0.8% agarose gel. The expression levels of 12 *Cr**Hsf*s were determined by qRT-PCR, and three biological replications for each treatment were conducted. In brief, a total of 1 µg of RNA was reverse transcribed into cDNA in a 20 µL reaction volume using AMV reverse transcriptase (CISTRO BIO, Guangzhou, China), according to the supplier’s instructions. To quantify the relative transcript levels of selected *CrHsf* genes, qRT-PCR was performed with gene-specific primers using a LightCycler480 system (Roche, Basel, Switzerland) and 2× Ultra SYBR Green qPCR Mix (CISTRO BIO, Guangzhou, China), according to the manufacturer’s instructions. The gene-specific primers used for this analysis are listed in Appendix A. All gene expression data obtained via qRT-PCR were normalized to the expression of *CrEF-1α* (Appendix A).

### 4.7. Cloning of CrHsf cDNAs and Transcriptional Activity Analysis of CrHsfs

Based on the RNA-Seq data, the full-length open reading frame of eight *CrHsf* A class members, including *CrHsf1*, *CrHsf2*, *CrHsf7*, *CrHsf9*, *CrHsf10*, *CrHsf15*, *CrHsf17*, and *CrHsf23*, were amplified from cDNA that was reverse transcribed from the total RNA of *C. rosea* seedling plants. Then, the PCR fragments were inserted into the *Eco*RI site of the pGBKT7 vector to generate *CrHsf*s-pGBKT7 in-frame fusion proteins, following the in-fusion cloning technique of the In-Fusion HD^®^ Cloning System (Takara/Clontech, Mountain View, CA, USA). After sequencing confirmation, the fusion constructs and control pGBKT7 (negative control) were transformed into yeast strain AH109 using the LiOAc/PEG method. The yeast clones were cultured in liquid synthetically defined medium (SD/-Trp) to OD600 until 2.0, after which they were diluted using a gradient dilution (1:10, 1:100, and 1:1000). Two-microliter yeast cultures were spotted onto the corresponding synthetically defined (SD/-Trp and SD/-Trp/-His) medium plates for 2 days at 30 °C. Yeast transformation and determination of blue/white colonies were conducted according to the instructions of the manufacturer (Clontech, CA, USA), and X-α-Gal was used as a substrate for the reporter gene MEL1. Primers used for plasmid construction are shown in Appendix A.

The transient expression assay of LUC reporter gene was performed in tobacco leaf lower epidermis cells with injection of *Agrobacterium tumefaciens* GV3101. In brief, the coding sequences of eight *CrHsf*s were amplified by PCR, verified by DNA sequencing, and cloned into pBIm, a modified binary vector of pBI121 without *GUS* report gene, to generate overexpression constructs. The corresponding primers are listed in Appendix A. The reporter plasmids (4 × GAAACTTC (HSE)-mini35S-pGreenII0800-LUC or 4 × GACACACT (mHSE)-mini35S-pGreenII0800-LUC) were kindly provided by Wei et al. [52], and transferred into GV3101:pSOUP. The expression vectors pBIm-*CrHsf*s and negative control pBI-GFP were transferred into GV3101. The GV3101:pSOUP culture containing the reporter vectors of either 4 × HSE-mini35S-pGreenII0800-LUC or 4 × mHSE-mini35S-pGreenII0800-LUC were mixed with the GV3101 containing expression vectors (volume ratio as 1:1), then, the mixed GV3101 cultures were centrifuged and dispersed in the infection solution (10 mM MES, 10 mM MgCl_2_, 150 mM acetosyringone, pH 5.6) until OD600 values of about 0.75. These GV3101 solutions were injected into the the lower epidermis of tobacco leaves. The tobacco seedlings were cultivated for a further 2 to 3 days and the leaves were captured and analyzed after the initial photographing of the tobacco leaves sprayed with D-Luciferin (10 µM) using a ChemiDOC^TM^ MP Imaging System (Bio-Rad, Hercules, CA, USA). The relative dual-luciferase (LUC/REN) activities were determined using a Dual-LUC Reporter Assay System E2920 (Promega, Madison, WI, USA). Three replicates were measured.

### 4.8. Functional Identification of CrHsfs in Yeast

The full coding regions of eight *CrHsf* A class members were amplified from cDNA that was reverse transcribed from the total RNA of plant leaves. The specific primers used for yeast expression vector construction are presented in Appendix A. The PCR products were then inserted into *Bam*HI and *Eco*RI sites of the pYES2 vector using In-Fusion^®^ techniques (Clontech, CA, USA) to yield recombinant plasmids *CrHsf*s-pYES2. After sequencing confirmation, these constructs, combined with empty vector pYES2, were transformed into different yeast strains using the standard LiAc/PEG method.

In this study, the yeast WT strain BY4741 (Y00000) and two deletion mutants, *yap1∆* (Y50569) and *skn7∆* (Y02900), were obtained from Euroscarf (http://www.euroscarf.de, accessed on 1 May 2022). A solid synthetic drop-out (SD) uracil medium with 2% galactose (SDG/-U) was used for the selection of yeast transformants. Subsequently, single clones possessing the empty vector pYES2 (control) or recombinant plasmids, *CrHsf*s-pYES2, were selected and inoculated in liquid SDG/-U medium overnight or longer at 30 °C. Then, the yeast solution was transferred into fresh liquid SDG/-U medium with a volume ratio of 1:100 and cultured for another 24 h or longer at 30 °C (150–200 rpm) until an OD600 of 2.0. The spot assays were performed on SDG plates under different extents or concentrations of stress (H_2_O_2_ for oxidative stress, 52 °C for heat stress, NaCl for salt stress, and sorbitol for high osmotic stress) indicated in the figure legends. The test plates were incubated at 30 °C for 2–5 days, and the pictures were taken based on the growth status of the yeast spots.

### 4.9. Statistical Analysis

All the experiments in this study were repeated three times independently, with the results shown as the mean ± SD (n ≥ 3). Pairwise differences between means were analyzed using Student’s *t*-tests in Excel 2010 (Microsoft Corporation, Albuquerque, NM, USA).

## 5. Conclusions and Prospects

Heat is a major abiotic stress factor that *C. rosea* plants have to adequately deal with in their full lifetime at their native habitats. In this study, the *CrHsf* gene family from *C. rosea* was systematically identified and analyzed, mainly concerning their genomic structures, segmental duplications, multiple alignment, motif analysis, and phylogenetic comparison. Based on the *C. rosea* genome data, 28 *CrHsf* genes were investigated by their evolutionary origin and phylogenetic relationship, other characteristics concerning environmental adaptation, including expression profiles and transcriptional regulatory mechanisms, were also fully explored. The expression profiles of *CrHsf*s suggested that the majority of *CrHsf*s may function in the growth and development of *C. rosea* plants. Transcriptional analyses also indicated that this gene family was involved in the adaptation of *C. rosea* to tropical coral reefs, as well as responding to salt/alkaline stress, high osmotic stress, or heat shock treatment. Transgenic yeast strains overexpressing several *CrHsf*s enhanced tolerance to heat stress and H_2_O_2_, and also showed transcriptional activation activities, which was consistent with their predicted roles as TFs that upregulate heat stress proteins or ROS scavenger genes. Overall, the bioinformatic analyses and expression profile studies of *CrHsf*s are helpful in understanding the important role of *Hsf*s in *C. rosea* in ecological adaptation to tropical coral islands and also provide the foundation for exploring methods to understand and regulate these stress responses. The results provide a foundation for subsequent exploration of leguminous plant *Hsf* gene functions. In addition, comprehensive characterization of multifunctional *CrHsf*s would provide the basis for investigating their possible roles in plant abiotic stress responses.

## Figures and Tables

**Figure 1 ijms-23-12357-f001:**
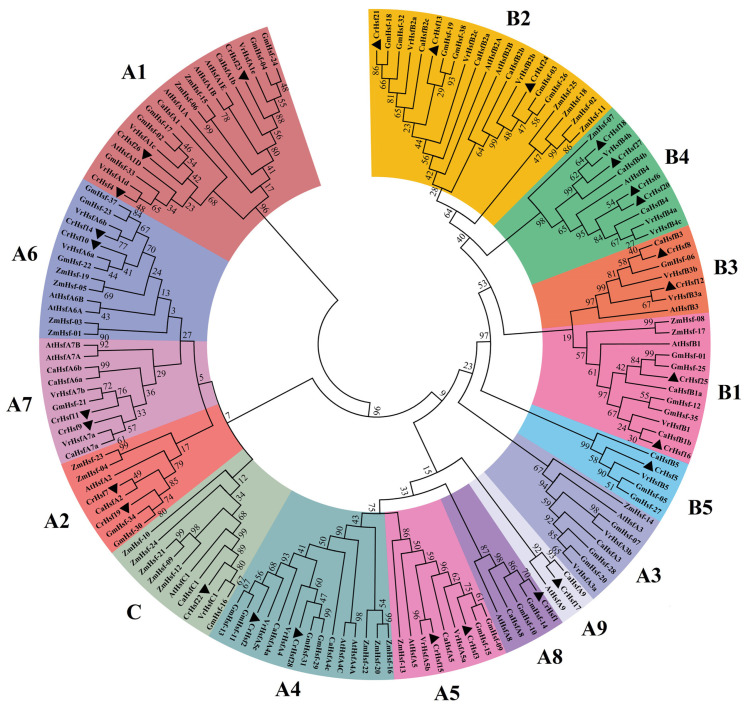
Phylogenetic relationships of the 28 CrHsfs from *Canavalia rosea*, the 38 GmHsfs from soybean (*Glycine max*), the 22 CaHsfs from chickpea (*Cicer arietinum*), the 24 VrHsf from mung bean (*Vigna radiata*), the 25 ZmHsfs from maize (*Zea mays*), and the 21 AtHsfs from Arabidopsis (*Arabidopsis thaliana*). The phylogenetic tree is constructed using MEGA 6.0 software with the Hsf protein sequences, by ClustalW alignment, the neighbor-joining (NJ) method, the bootstrap method, and 1000 repetitions. All subclasses (A1–A9, B1–B5, and C) of Hsf proteins are well separated in different clades and represented by different color backgrounds.

**Figure 2 ijms-23-12357-f002:**
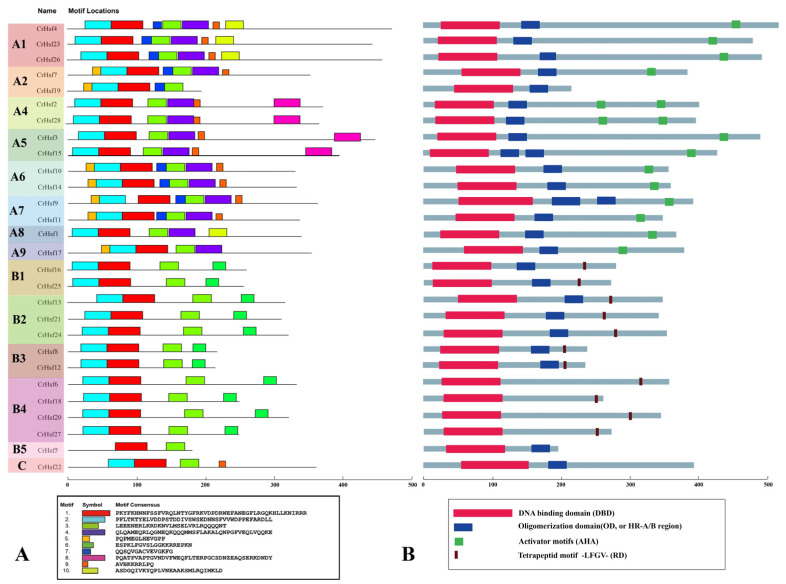
The motif composition predicted by the MEME prediction (http://meme-suite.org/, accessed on 1 May 2022) (**A**) and the conserved domain diagram drawn manually (**B**) of *C. rosea* Hsf proteins. The far left indicates the constructed phylogenetic tree with MEGA 6.0, including group A (A1, A2, A4, A5, A6, A7, A8, A9), group B (B1, B2, B3, B4, B5), and group C. The ten conserved motifs are listed in the left bottom box, and the four functional domains, including DBD, OD, AHA, and RD, are listed in the right bottom box.

**Figure 3 ijms-23-12357-f003:**
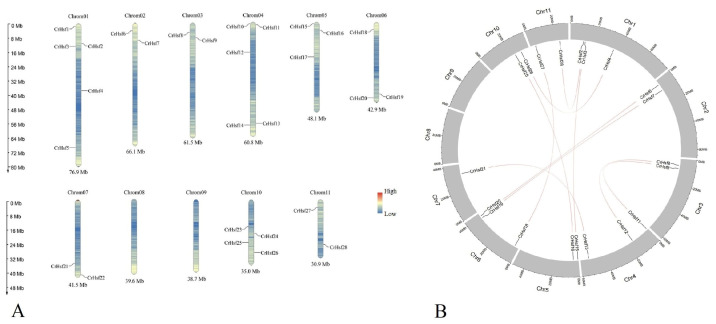
(**A**) Chromosomal distribution of 28 *CrHsf* genes in the *C. rosea* genome. The scale of the chromosome is showed in millions of bases (Mb); (**B**) the distribution of segmental duplication of Cr*Hsf*s in *C. rosea* chromosomes.

**Figure 4 ijms-23-12357-f004:**
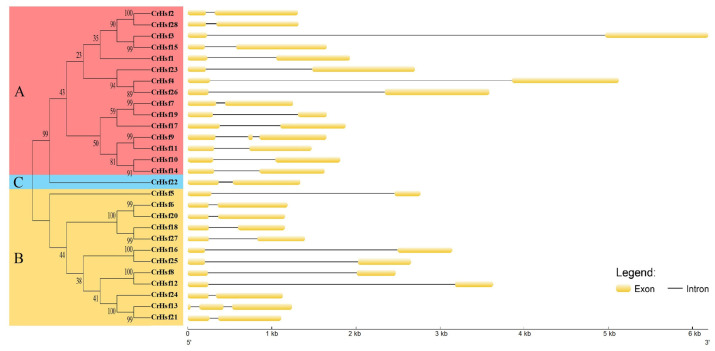
The exon-intron organization of the *CrHsf* genes constructed with GSDS 2.0 (http://gsds.cbi.pku.edu.cn/, accessed on 1 May 2022). The left capital letters “A, B, and C” and corresponding phylogenetic tree represent different groups marked with different colors (red for Group A, yellow for Group B, and blue for Group C).

**Figure 5 ijms-23-12357-f005:**
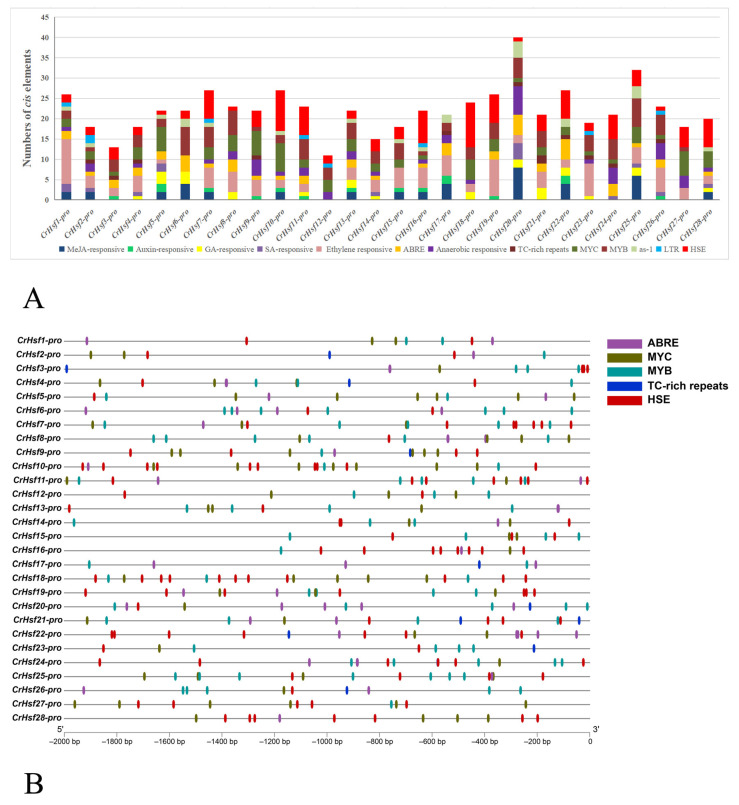
Prediction of *cis*-regulatory elements in the 2000 bp upstream regulatory regions of *CrHsf* genes: (**A**) Summaries of the thirteen *cis*-regulatory elements in the 28 *CrHsf* promoter regions; (**B**) distribution of the five *cis*-regulatory elements (ABRE, MYC, MYB, TC-rich repeat, and HSE) in the *CrHsf* genes promoter regions. The scale bar represents 200 bp.

**Figure 6 ijms-23-12357-f006:**
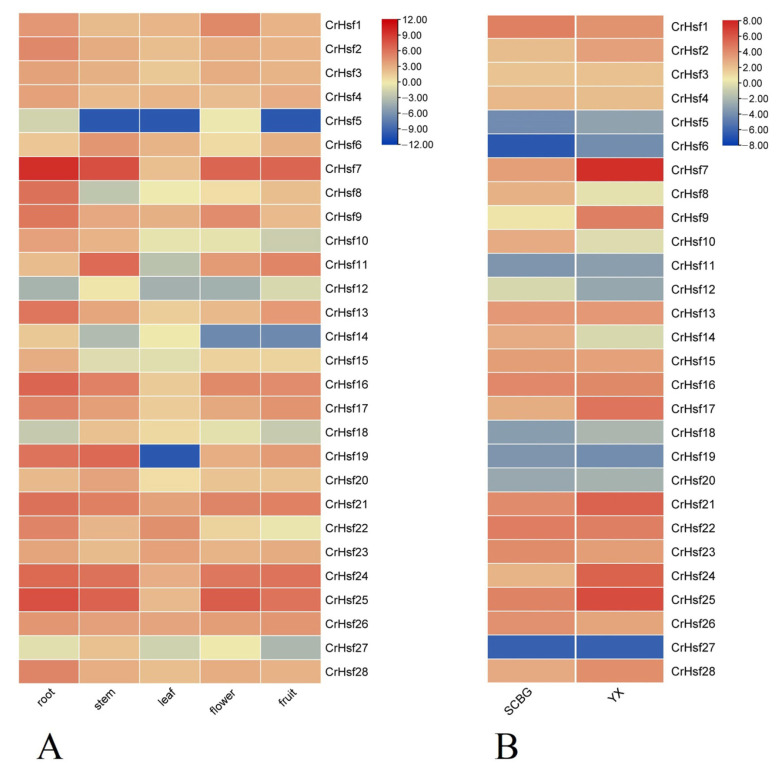
Heatmaps showing: (**A**) The expression levels of the 28 *CrHsf* genes in the root, stem, leaf, flower bud, and young fruit of *C. rosea* plant; (**B**) expression differences of the 28 *CrHsf* genes in mature *C. rosea* leaves captured from adult plants growing in the South China Botanical Garden (SCBG) and on Yongxing Island (YX). The expression level of each gene is shown in FPKM values (log2). Orange red denotes high expression levels, and dark blue denotes low expression levels.

**Figure 7 ijms-23-12357-f007:**
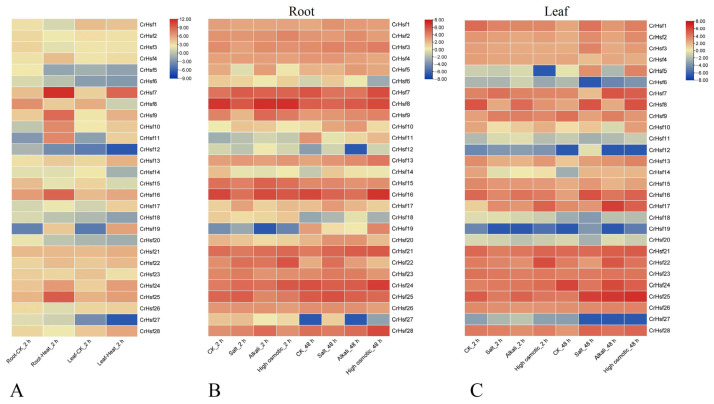
Heatmaps showing: (**A**) The expression levels of the 28 *CrHsf* genes in roots and leaves of *C. rosea* seedlings under heat shock stress (45 °C for 2 h); the expression patterns of the 28 *CrHsf* genes under abiotic stress treatment (600 mM NaCl, 150 mM NaHCO_3_, pH 8.2, 300 mM mannitol, for 2 h and 48 h) in roots (**B**) and leaves (**C**). The expression level of each gene is shown in FPKM values (log2). Orange red denotes high expression levels, and dark blue denotes low expression levels.

**Figure 8 ijms-23-12357-f008:**
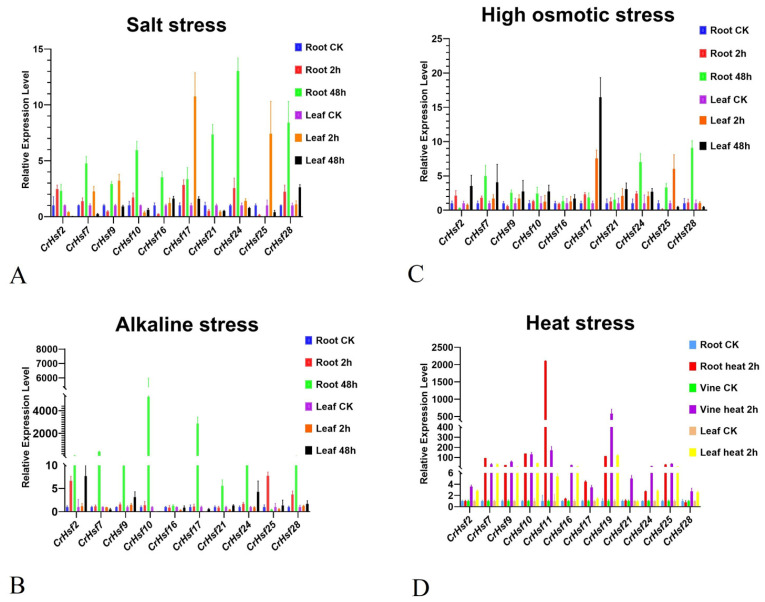
Tempo-spatial expression analysis of the 10 *CrHsf* genes with quantitative reverse transcription PCR (qRT-PCR) responding to different stresses: (**A**) The relative expression patterns of 10 *CrHsf* genes under salt stress (150 mM NaCl); (**B**)alkaline stress (NaHCO_3_); (**C**) high osmotic stress (300 mM mannitol); (**D**) heat stress (45 °C), in *C. rosea* seedling plants. Relative expression values were calculated using the 2^−ΔCt^ method with the housekeeping gene *CrEF-1α* as the reference gene.

**Figure 9 ijms-23-12357-f009:**
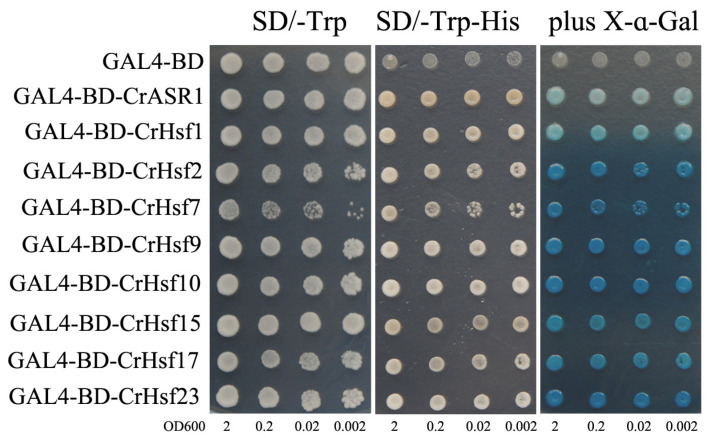
Transactivation assay of the eight CrHsf proteins in yeast cells. The GAL4 DNA binding domain was fused with eight CrHsfs and transformed into the yeast strain AH109 containing the *His3* and *LacZ* reporter genes. An analysis of β-galactosidase activity of the relative yeast strains on plates was also performed. The yeast culture (OD600 to 2) was serially diluted to OD600 values of 0.2, 0.02, and 0.002, and then the 2 μL yeast liquid was spotted onto the SD plates and cultured for 2 d at 30 °C. Negative control, yeast cells transformed with pGBKT7 empty vector; positive control, yeast cells transformed with CrASR1-pGBKT7 [25]. Experiments were performed three times with similar results.

**Figure 10 ijms-23-12357-f010:**
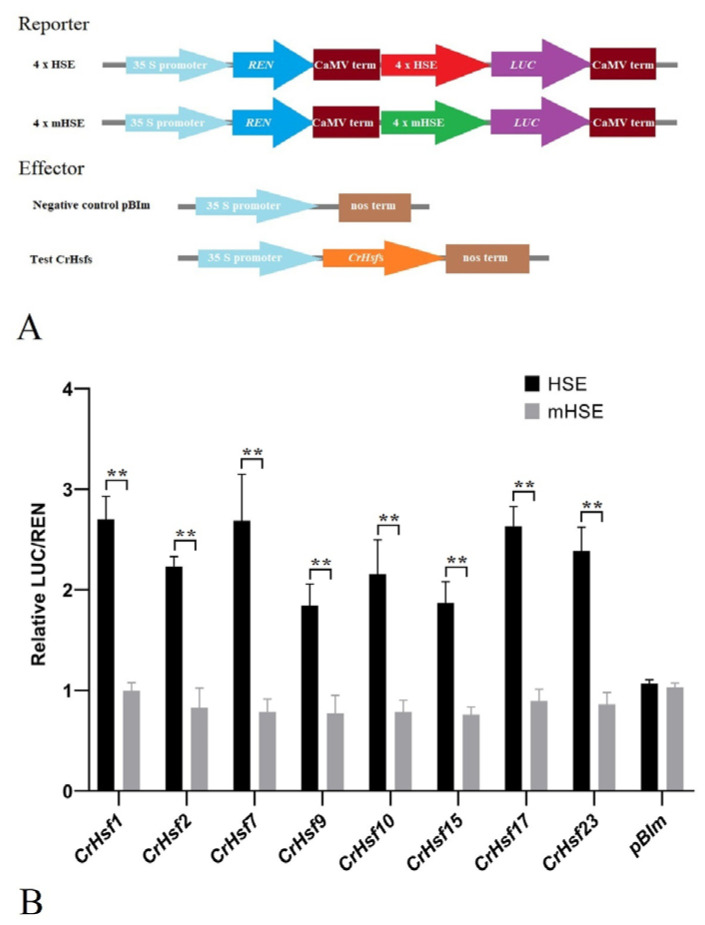
The dual-luciferase assay of the eight CrHsf proteins on synthetic reporter plasmids (4 × GAAACTTC (HSE)-pGreenII0800-LUC or 4 × GACACACT (mHSE)-pGreenII0800-LUC): (**A**) The schematic diagram of the constructs. REN, Renilla luciferase; LUC, firefly luciferase; terM, terminator; (**B**) the ratio of LUC/REN activity in *Nicotiana benthamiana* mesophyll cells co-transfected with agrobacterium GV3101 containing different LUC reporters (HSE or mHSE) and REN effectors (different CrHsfs and pBIm for testing or empty vector pBIm as negative control). ** The values are means ± SD of three biological replicates.

**Figure 11 ijms-23-12357-f011:**
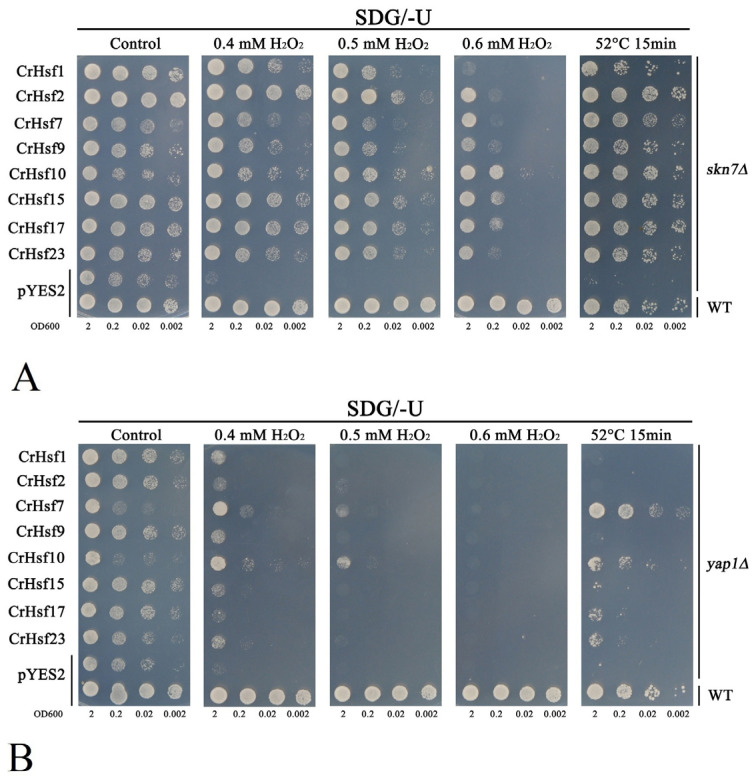
The tolerance confirmations of the eight *CrHsf* gene heteroexpression in yeast. H_2_O_2_ oxidative stress and heat stress (52 °C 15 min) tolerance confirmations in yeast mutant strain *skn7∆* (**A**) and *yap1∆* (**B**). The stress factors with different concentrations are shown in the figures. Yeast cultures were adjusted to OD600 = 2, and 2 μL of serial dilutions (10-fold, from left to right in each panel) was spotted on SDG/-Ura medium supplemented with different concentrations of H_2_O_2_ (0.4 mM, 0.5 mM, and 0.6 mM), and small portions of the yeast cultures were incubated at 52 °C for 15 min, and then were moved to a 30 °C environment before being spotted on SDG/-Ura medium (without any chemical stress factors) for the thermotolerance confirmation. Corresponding yeast spots growing on SDG/-Ura plates without H_2_O_2_ or heat stress were used as the control. The plates were incubated for 2–5 days at 30 °C. The images are representative of three independent experiments.

**Figure 12 ijms-23-12357-f012:**
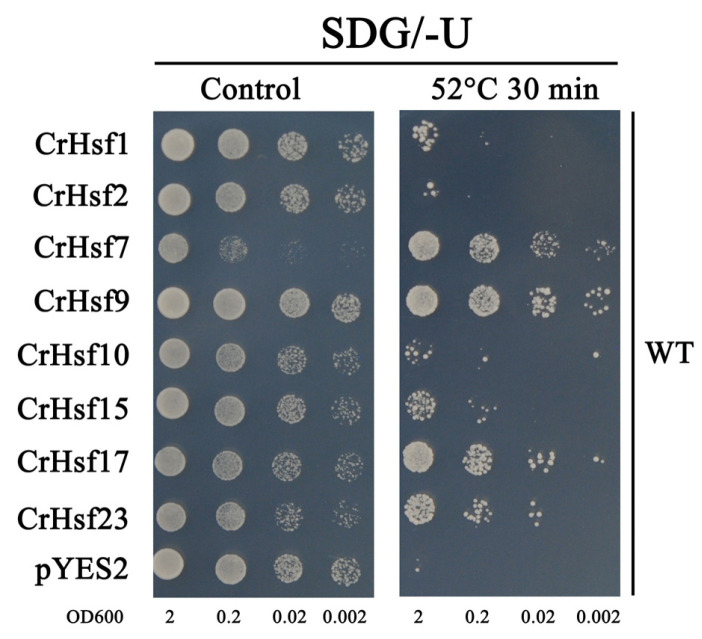
The heat stress tolerance confirmation (52 °C 30 min) of the eight *CrHsf* genes heteroexpression in yeast wild type strain (WT). Yeast cultures were adjusted to OD600 = 2, and small portions of the yeast cultures were incubated at 52 °C for 30 min, and then were moved to a 30 °C environment before being spotted on SDG/-Ura medium (without any chemical stress factors) for the thermotolerance confirmation. The corresponding yeast spots growing on SDG/-Ura plates without heat stress were used as the control. The plates were incubated for 2–5 days at 30 °C. The images are representative of three independent experiments.

**Table 1 ijms-23-12357-t001:** Nomenclature of heat shock transcription factor genes and subcellular localization of the coding proteins identified from the *Canavalia rosea* genome.

Name	Gene Locus	Class	CDS (bp)	Mw (kDa)	PI	II	GRAVY	Subcellular Localization (WoLF_PSORT)
CrHsf1	01T000116	A8	1104	41.85	4.9	51.75	−0.678	nucl: 13.5, cyto_nucl: 7.5
CrHsf2	01T000944	A4	1209	45.78	4.9	55.10	−0.752	nucl: 14
CrHsf3	01T001060	A5	1452	54.53	5.64	61.31	−0.777	nucl: 14
CrHsf4	01T002065	A1	1533	56.16	4.89	58.37	−0.637	nucl: 14
CrHsf5	01T002937	B5	588	22.39	8.23	56.00	−0.598	nucl: 13, pero: 1
CrHsf6	02T004235	B4	1080	40.28	8.62	47.44	−0.599	nucl: 14
CrHsf7	02T004720	A2	1146	42.50	5.03	66.71	−0.541	nucl: 13.5, cyto_nucl: 7.5
CrHsf8	03T007873	B3	705	26.76	8.85	58.96	−0.828	nucl: 13, pero: 1
CrHsf9	03T008076	A7	1179	44.72	5.23	61.06	−0.669	cyto: 4.5, E.R.: 3, cyto_pero: 3, nucl: 2, vacu: 2, mito: 1, plas: 1
CrHsf10	04T011078	A6	1074	41.35	4.76	62.10	−0.791	nucl: 14
CrHsf11	04T011116	A7	1047	40.66	5.58	56.11	−0.798	nucl: 14
CrHsf12	04T011879	B3	699	26.88	8.82	52.24	−0.711	nucl: 14
CrHsf13	04T013612	B2	1029	38.46	8.98	52.93	−0.720	nucl: 14
CrHsf14	04T013679	A6	1080	41.63	5.08	49.36	−0.710	nucl: 14
CrHsf15	05T014277	A5	1281	48.70	5.34	63.89	−0.854	nucl: 14
CrHsf16	05T014583	B1	849	30.92	5.61	42.86	−0.668	nucl: 13, pero: 1
CrHsf17	05T015720	A9	1152	43.64	5.88	42.28	−0.712	nucl: 14
CrHsf18	06T017427	B4	810	31.30	6.63	55.13	−0.639	nucl: 14
CrHsf19	06T018996	A2	636	23.78	6.12	42.87	−0.643	nucl: 14
CrHsf20	06T019137	B4	1044	39.60	7.73	51.42	−0.710	nucl: 14
CrHsf21	07T020710	B2	1008	37.00	6.27	61.01	−0.645	nucl: 14
CrHsf22	07T021167	C	1173	43.35	6.77	48.59	−0.608	nucl: 12.5, cyto_nucl: 7, chlo: 1
CrHsf23	10T026034	A1	1440	53.42	5.35	55.90	−0.556	nucl: 14
CrHsf24	10T026239	B2	1044	37.90	5.16	53.51	−0.517	nucl: 14
CrHsf25	10T026586	B1	834	30.97	8.98	39.60	−0.928	nucl: 11, cyto: 1, mito: 1, plas: 1
CrHsf26	10T027023	A1	1488	54.79	4.99	64.79	−0.678	nucl: 14
CrHsf27	11T027981	B4	810	31.43	7.10	59.72	−0.622	nucl: 14
CrHsf28	11T029135	A4	1197	45.65	5.06	52.51	−0.764	nucl: 14

CDS: coding DNA sequence of *CrHsf* genes. The physicochemical parameters, including molecular weight (Mw, kDa) and theoretical isoelectric point (PI), of each CrHsf protein were calculated using the compute PI/Mw tool of ExPASy (http://www.expasy.org/tools/, accessed on 1 May 2022). GRAVY (grand average of hydropathy) values were calculated using the PROTPARAM tool (http://web.expasy.org/protparam/, accessed on 1 May 2022). Subcellular location prediction of CrHsf proteins was conducted using the WoLF_PSORT (https://www.genscript.com/wolf-psort.html, accessed on 1 May 2022). Nucl, nucleus; plas, plasma membrane; cyto, cytoplasm; mito, mitochondria; ER, endoplasmic reticulum; pero, peroxisomal; chlo, chloroplast; golg, golgi body; vacu, tonoplast membrane; cysk, cytoskeleton; extr, extracellular region; secr, secretory. The scores represent the probabilities of subcellular localization.

**Table 2 ijms-23-12357-t002:** Functional domains of CrHsfs.

Protein-Class	Length	DBD	HR-A/B	NLS	NES	AHA	RD
CrHsf4-A1	510 aa	30–119	148–175	N.D.	N.D.	(447)DLFNNPLFWD	N.D.
CrHsf23-A1	479 aa	23–103	130–157	(215)ITGGNKKRRLHRQ	N.D.	(414)DEFWELFFMP	N.D.
CrHsf26-A1	495 aa	24–113	174–190	N.D.	N.D.	(431)DDFLSNPSIW	N.D.
CrHsf7-A2	381 aa	54–143	170–190	(141)LLKTIKRRRNVT, (250)VRRKRRLTAS	N.D.	(320)GSVWEDLLNQ (360)DDDWTEDLQS	N.D.
CrHsf19-A2	211 aa	41–130	158–178	N.D.	N.D.	N.D.	N.D.
CrHsf2-A4	402 aa	14–103	125–152	N.D.	(142)L	(254)VAFWEAIAQD (338)DVFWEQFLTE	N.D.
CrHsf28-A4	398 aa	14–103	120–175	(208)VDRKRRLPRS	(134)LEKLKHEKEQL	(256)MAFWENLARD (340)DVFWEQFLTE	N.D.
CrHsf3-A5	483 aa	18–107	125–145	(208)LSAYNKKRRLPQV	(203)L	(430)DVFWEQFLTE	N.D.
CrHsf15-A5	426 aa	9–98	116–136, 151–171	(199)LSAYNKKRRLPQV	(307)L	(386)DMFWEQFLTE	N.D.
CrHsf10-A6	357 aa	43–132	159–200	(240)LCKKRRRPID	(285)L, (342)L	(317)EVFWEDFLNE	N.D.
CrHsf14-A6	359 aa	46–135	164–205	(245)FSKKRRRPID	(284)LEFEVDL	(322)EVFWQNLLNE	N.D.
CrHsf9-A7	392 aa	51–160	189–222, 251–271	(270)MSKKRRRPIE	(197)LVL	(348)EGFWEELFSE	N.D.
CrHsf11-A7	348 aa	46–135	159–179	(239)LTKKRRRQIE	(279)L, (281)M	(306)EQFWEEVLFG	N.D.
CrHsf1-A8	367 aa	19–108	143–170	N.D.	(178)LQ	(317)SPFLGNVQDS	N.D.
CrHsf17-A9	383 aa	68–157	168–195	(231)RMARKPAFVEQLIQKIKRKRELDGNDMDKRPRL	(189)I, (192)L	(278)QGFQSELNGL	N.D.
CrHsf16-B1	282 aa	10–99	137–171	(260)NCRKRGRQDPIAAGAKQLKT	(281)I	N.D.	234
CrHsf25-B1	277 aa	10–99	160–180	N.D.	(185)IAFLKERL	N.D.	222
CrHsf13-B2	342 aa	48–137	202–222	N.D.	(212)L	N.D.	278
CrHsf21-B2	335 aa	28–117	182–202	N.D.	(192)L, (201)SL	N.D.	265
CrHsf24-B2	347 aa	25–114	180–207	N.D.	(190)L, (195)M, (197)L	N.D.	281
CrHsf8-B3	234 aa	22–111	154–174	(174)TNMKRKCREL, (215)GGRDMKRNRAE	N.D.	N.D.	201
CrHsf12-B3	232 aa	23–112	177–197	(176)TIMKRKCREL, (215)GEREMKKRRDEI	N.D.	N.D.	201
CrHsf6-B4	359 aa	25–114	N.D.	N.D.	N.D.	N.D.	312
CrHsf18-B4	269 aa	26–115	N.D.	N.D.	N.D.	N.D.	249
CrHsf20-B4	347 aa	25–114	N.D.	N.D.	(338)L	N.D.	299
CrHsf27-B4	269 aa	26–115	N.D.	N.D.	N.D.	N.D.	252
CrHsf5-B5	195 aa	31–124	152–182	N.D.	(167)LELQM	N.D.	N.D.
CrHsf22-C	390 aa	65–154	174–208	N.D.	(184)LKEEQKALEEEL	N.D.	N.D.

DBD, DNA-binding domain (PF00447); HR-A/B, also called oligomerization domain (OD, heptad hydrophobic repeat A or B, coiled-coil structure); NLS, nuclear localization signal; NES, nuclear export signal; AHA (aromatic “W/F/Y”, larger hydrophobic “L/I/V”, and acidic “E/D” amino acid residues), transcriptional activation domain; RD, repressor domain (high concerved tetrapeptide -LFGV-); N.D., no motifs detectable by sequence similarity search. For the NLS and NES columns, the numbers in parenthesis are the start sites of the functional domain. Pfam (http://pfam.xfam.org/search, accessed on 1 May 2022), cNLS Mapper (http://nls-mapper.iab.keio.ac.jp/cgi-bin/NLS_Mapper_form.cgi, score ≥ 5.0. accessed on 1 May 2022), and NetNES (http://www.cbs.dtu.dk/services/NetNES/, score > 0.5. accessed on 1 May 2022) were used to identify these conserved domains, including DBD, HR-A/B, NLS, and NES domains, respectively. The AHA and RD motifs were predicted manually according to previous reports [11,18,19].

**Table 3 ijms-23-12357-t003:** Ka/Ks analysis (NG method) and duplicated data calculations for *CrHsf* genes.

Duplicated Pair	Subfamily	Duplicate Type	Ka	Ks	Ka/Ks	*p*-Value (Fisher)	Positive Selection
*CrHsf4*/*CrHsf26*	A1	Segmental	0.130461	0.519849	0.251	6.11 × 10^−24^	No
*CrHsf7*/*CrHsf19*	A2	Segmental	0.183841	1.25915	0.146	8.24 × 10^−24^	No
*CrHsf2*/*CrHsf28*	A4	Segmental	0.149982	0.613725	0.2444	4.17 × 10^−21^	No
*CrHsf3*/*CrHsf15*	A5	Segmental	0.169137	0.527677	0.3205	2.42 × 10^−15^	No
*CrHsf9*/*CrHsf11*	A7	Segmental	0.181549	0.581786	0.3121	8.48 × 10^−14^	No
*CrHsf16*/*CrHsf25*	B1	Segmental	0.257367	1.21105	0.2125	8.40 × 10^−21^	No
*CrHsf13*/*CrHsf21*	B2	Segmental	0.159903	0.969807	0.1649	9.10 × 10^−32^	No
*CrHsf8*/*CrHsf12*	B3	Segmental	0.140623	0.825046	0.1704	2.16 × 10^−21^	No
*CrHsf6*/*CrHsf20*	B4	Segmental	0.179072	0.98243	0.1823	4.90 × 10^−30^	No
*CrHsf18*/*CrHsf27*	B4	Segmental	0.101033	0.532705	0.1897	5.81 × 10^−17^	No

## Data Availability

Not applicable.

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
