# Peer review of "Functional Characterization of Heat Shock Factor (CrHsf) Families Provide Comprehensive Insight into the Adaptive Mechanisms of Canavalia rosea (Sw.) DC. to Tropical Coral Islands"

_ijms, 2022, doi:10.3390/ijms232012357_

Round 1

Reviewer 1 Report

I have reviewed the manuscript entitled Functional characterization of heat shock factors (CrHsfs) fam- 2 ily provides comprehensive insight into the adaptive mecha- 3 nisms of Canavalia rosea (Sw.) DC. to tropical coral islands.

The introduction is well written. The two major concerns I would raise are that all figures are of poor quality, making it difficult for the reader to make out the text and what they actually show. The authors should make the figures larger and in better resolution.

The second comment about the manuscript is that the authors describe heat stress. The introduction does not mention that heat stress is tested on roots. Normally, when we hear the term heat stress, we think of high air temperatures around the plant. Roots usually grow in soil that is cooler than the air itself. In the Materials and Methods, the authors state that the water was set at 45 °C and the roots were exposed to this temperature. So the authors exposed the roots to a heat shock, taking into account that the irrigation water or other water sources have a temperature of 45 °C?

This concerns should be explained and addresed by the authors.

Author Response

We really appreciate your confirmation!

  1. About the poor-quality figures

We are sorry that we uploaded the compressed figure files at the beginning of this submission. Here we also submitted the original figures within the attached files.

  1. The high temperatures challenges to roots of roseaseedlings

We agree with the opinion that in normal habitat, the roots of plants will not withstand high temperature stress, since they are deep in soil. But things are absolutely different on the tropical coral reefs, under the direct sunlight at the noon, the coral sand became hot and blazing. Actually for the seedlings of sea-floating plants such as C. rosea, their roots were not buried so deep in the sand at their native habitats. Sometimes the temperature of coral sand surface on tropical islands could reach 50‒60℃. Here we think the 45℃ heat challenge for both shoot and root of C. rosea seedling were suitable and usual for this special habitat species.

Reviewer 2 Report

The Authors presented a well-written, valuable manuscript on the heat shock factor gene family of C. rosea. I recommend the publication of the manuscript in International Journal of Molecular Sciences.

Author Response

We gratefully appreciate for your approval!

Reviewer 3 Report

The manuscript entitled "Functional characterization of heat shock factors (CrHsfs) family provides comprehensive insight into the adaptive mechanisms of Canavalia rosea (Sw.) DC. to tropical coral islands" reports the multifunctionality of the CrHsf gene which is involved in the natural ecological adaptability of C. rosea to abiotic stress. Their data were well supported by appropriate analyses for Hsf gene functions. The work was well done according to the procedures. The investigations of this research are interesting.

Author Response

We are very grateful for your approval. We have modified some expression in this manuscript to make it more precise and fluent. And these revisions were marked in red.

Reviewer 4 Report

The manuscript, "Functional characterization of heat shock factors (CrHsfs) family provides comprehensive insight into the adaptive mechanisms of Canavalia rosea (Sw.) DC. to tropical coral islands " from Mei Zhang, Zhengfeng Wang and Shuguang Jian presents a genome-wide analysis of C. rosea Hsfs, their expression levels and functional identification.

The manuscript, overall, is well written, and the results report a carefully designed analysis of the aim of the study.  However, the manuscript needs minor revision as specified within the attached pdf file. The authors will find some specific points for revision in yellow marker with comments. Some figures (e.g., Figure 5A) need better resolution. In addition, some sentences in the Results section, in green marker, could be included in the discussion.

Finally, I would like to point out that the authors did not send the additional files mentioned in the manuscript.

Author Response

  1. The aim of the study is not so clear

    Canavalia rosea is a typical sea-drifted leguminous halophyte with good adaptability to tropical coral islands or reefs, in which often possessing the characteristic of high temperature, high alkaline, high salinity, high light, fresh water shortage, and soil fertility shortage. At the beginning of this research, we just thought about the salt and drought tolerance of C. rosea being a tropical halophyte. Along with research going deep, we found the compared with other halophytic plant species, the thermotolerance of C. rosea might be a prerequisite for its special ecological adaptability to the tropical coral reef. Then, the Hsf genes’ functions will be not insignificant.

    The multifunctionality of plant Hsf genes have been confirmed in many other plant species, which mainly includes mediating the complex signaling systems of high temperature challenges, as well as other stress responses, such as oxidative stress, drought, hypoxic conditions, soil salinity, toxic minerals, strong irradiation, even pathogen threats. Plant Hsfs could up-regulate many stress-inducible genes, including protective HSPs and other chaperones, ROS scavengers, enzymes, apoptotic regulators, or some other TFs. Their functional diversity has been summarized by N Andrási et al (J Exp Bot, 2021. doi:10.1093/jxb/eraa576). Here in this study we reinforced the thermotolerance mediated by the CrHsfs in C. rosea plants, and we also mentioned the salinity/alkaline and drought tolerance possibly involved in the CrHsfs’ network. We also performed some transgenic assay in plants, and we are going to continue to explore the genes’ functions, mainly focused on the molecular mechanisms of ecological adaptation to tropical coral islands in C. rosea.

  1. About the additional files

    Thank you so much for your careful check, and we have made correction according to the Reviewer’s comments. They were marked in red in the revised manuscript. And we also uploaded all of the full Supplementary material files and high-resolution pictures of this manuscript.

Round 2

Reviewer 1 Report

I thank the authors for the explanations.